# A Comparison of the Sensitivity of Contrast-Specific Imaging Modes on Clinical and Preclinical Ultrasound Scanners

Carmel M. Moran [1], Charles Arthur [1] and Emilio Quaia [1,2,*]

1  Centre for Cardiovascular Science, Queen's Medical Research Institute, University of Edinburgh, Edinburgh EH16 4TJ, UK
2  Department of Radiology, University of Padua, 35128 Padua, Italy
*  Correspondence: emilio.quaia@unipd.it; Tel.: +39-049-8212375

**Abstract:** Ultrasonic contrast agents are used routinely to aid clinical diagnosis. All premium- and mid-range scanners utilise contrast-specific imaging techniques to preferentially isolate and display the nonlinear signals generated from the microbubbles when insonated with a series of ultrasound pulses. In this manuscript the abilities of four premium ultrasound scanners to detect and display the ultrasound signal from two commercially available contrast agents—SonoVue and DEFINITY®—are compared. A flow phantom was built using tubes with internal diameters of 1.6 mm and 3.2 mm, suspended at depths of 1, 5 and 8 cm and embedded in tissue-mimicking material. Dilute solutions of SonoVue and DEFINITY® were pumped through the phantom at 0.25 mL/s and 1.5 mL/s. Four transducers were used to scan the tubes—a GE Logiq E9 (C2-9) curvilinear probe, a Philips iU22 L9-3 linear array probe, an Esaote MyLab Twice linear array LA523 (4–13 MHz) and a Fujifilm VisualSonics Vevo3100 MX250 (15–30 MHz) linear array probe. We defined a new parameter to compare the ability of the ultrasound scanners to display the contrast enhancement. This was defined as the ratio of grey-scale intensity ratio in contrast-specific imaging mode relative to the B-mode intensity from the same region-of-interest within the corresponding B-mode image. The study demonstrated that the flow rates used in this study had no effect on the contrast-specific imaging mode to B-mode (CSIM-BM) ratio for the three clinical scanners studied, with SonoVue demonstrating broadly similar CSIM-BM ratios across all 3 clinical scanners. DEFINITY® also displayed similar results to SonoVue except when insonated with the Esaote MyLab Twice LA523 transducer, where it demonstrated significantly higher CSIM-BM ratios at superficial depths.

**Keywords:** contrast; contrast-specific imaging; SonoVue; Definity; preclinical ultrasound

## 1. Introduction

Ultrasonic contrast imaging is now routinely used in most major radiology and cardiology clinical departments with all premium and most mid-range scanners utilising contrast-specific imaging software [1]. Guidelines and recommendations for the acquisition and interpretation of contrast-enhanced images to aid lesion detection and diagnosis in radiology and perfusion defects in cardiac applications are widely available [2–5]. Ultrasonic contrast-specific imaging modes (CSIM) are designed to detect, enhance and preferentially display the non-linear ultrasound signal generated by contrast microbubbles, while suppressing or cancelling the signals generated by tissue. CSIM techniques can be separated into high and low mechanical index (MI) techniques, where the mechanical index is defined as the quotient of the derated peak negative pressure to the square-root of the centre frequency of the transmitted wave. Detection mechanisms of contrast microbubbles capitalise on the non-linear behaviour of the bubbles when insonated at their resonant frequencies and at non-destructive, low mechanical index values (<0.1). At these low pressures, tissue predominantly exhibits linear scattering. To detect the non-linear contrast signal and suppress the linear tissue signal, a range of multi-pulse ultrasonic insonation

regimes have been developed—the most widely implemented are pulse/phase-inversion (PI) techniques [6], amplitude/power modulation (AM) techniques [7] or a combination of both techniques known as pulse inversion amplitude modulation (PIAM) [8]. In high mechanical index techniques (MI > 0.1), one or several short high-pressure pulses are emitted from the transducer to destroy the contrast microbubbles in the scan-plane. The subsequent refilling dynamics of contrast in the scan-plane imaged at lower mechanical index values, utilising techniques described above, and the use of maximum intensity projection algorithms, enables a picture of the vasculature architecture and perfusion-related indices to be investigated. This technique is called destruction-reperfusion or replenishment technique. A more detailed description of these techniques can be found elsewhere [9]. While ultrasound manufacturers have developed their own specific variations on these techniques and given them proprietary names, there have been limited studies to date that compare contrast-enhancement values on scanners optimised by radiologists for specific applications under a range of flow conditions and imaging-depths. Moreover, this information is of importance since clinical diagnosis is rarely dependent on linearising contrast image sequences and performing quantitative analysis but relies principally upon the clinician's or sonographer's visual assessment of the spatial and temporal enhancement pattern of contrast within an organ or vessel. Hence the set-up of the scanner optimised by the clinician is of importance. Consequently, the aim of this study was to measure the echo-enhancement of the DICOM images acquired from four premium ultrasound scanners and transducers using CSIM at different flow rates and depths and utilising two different contrast agents. Although for some scanners, it was possible to access images earlier in the processing path, analysis was performed on DICOM images downloaded from each of the clinical scanners as these are the images which are routinely used to determine the clinical diagnosis. The four scanners and transducers studied were a Philips iU22 L9-3 linear array probe (Philips Ultrasound, Bothell, WA, USA), an Esaote MyLab Twice Vet LA523 linear array probe (Esaote S.p.A, Genova, Italy) a GE Logiq E9 C2-9 curvilinear probe (GE, Boston, MA, USA) and a FujiFilm VisualSonics Vevo 3100 MX250 linear array probe (FujiFilm Visualsonics, Amsterdam, The Netherlands). Two commercially available clinical contrast agents DEFINITY® (Lantheus Medical Imaging, Inc. N Billerica, MA, USA) and SonoVue (Bracco Imaging SpA, Milan, Italy) were used in the study. Both contrast agents are composed of lipid-encapsulated microbubbles, SonoVue is composed of $SF_6$ microbubbles with a concentration $5.6 \times 10^8$ microbubbles/mL and DEFINITY® is composed of octafluoropropane microbubbles with a concentration of $6.4 \times 10^9$ microbubbles/mL. SonoVue is most widely used in Europe and China, while DEFINITY® dominates clinically in the USA.

## 2. Materials and Methods

A flow phantom was constructed within a perspex tank ($10 \times 11 \times 20$ cm) with 6 C-flex (Sanit-Gobain, Courbevoie, France) tubes suspended horizontally across the tank and threaded through engineered holes in the perspex ends. The inlet tubing was supported external to the Perspex tank on Perspex mouldings of length 10 cm to ensure laminar flow was well established within the section of the tubes that was scanned. To ensure laminar flow was established within the phantom, the minimum inlet length of the tube, L, had to be longer than:

$$L = 0.04 \times Re \times D \tag{1}$$

where Re is the Reynolds number and D is the internal diameter of the tube [10].

The Reynolds number is given by:

$$Re = \frac{\rho v D}{\mu} \tag{2}$$

where $\rho$ is equal to the density of the water ($1000$ kg m$^{-3}$), $\mu$ is the viscosity ($1$ mPa.s) and v is the velocity in m s$^{-1}$. Values for the minimum inlet lengths required are shown in Table 1.

Three tubes were of internal diameter (ID) 1.6 mm and the remaining three of ID 3.2 mm. Tubes of each diameter were centred at depths of 1, 5 and 8 cm from the surface of the tank (0.5, 4.5 and 7.5 cm from the surface of the tissue-mimicking material—TMM) and off-set from each other so that only one pipe was visible in the scanning plane. A system of three-way taps was attached to the end of each of the tubes to enable the contrast solution to be pumped into each of them. Once the tubes were in position, the tank was then filled with an IEC agar-based TMM [11,12], to a depth of 0.5 cm below the top of the tank. The TMM was then left to set overnight. The properties of this TMM are well documented by ourselves and others up to frequencies of 60 MHz [13,14]. After the TMM was set, TMM maintenance fluid, a combination of glycerol, water and antibacterial fluid (speed of sound 1540 m s$^{1}$), was poured onto the top of the phantom [15] and the phantom was covered by cling film to ensure the TMM did not dry out. The experiments described below were undertaken over a period of three months, during which there was no evidence of degradation of the phantom. All scanning experiments were undertaken in clinical radiology departments (GE Logiq E9), in University of Edinburgh veterinary radiology departments (Esaote MyLab Twice) or in biological research facilities (Philips iU22, FujiFilm Visualsonics Vevo 3100) in light levels similar to those used for clinical examinations (Figure 1).

**Table 1.** Calculated mean velocity at both flow settings and minimum inlet length to ensure laminar flow established within tubes in scanning plane.

| Flow Rate (cm$^3\cdot$s$^{-1}$) | 1.6 mm ID Tube [1] | 3.2 mm ID Pipe [1] |
|:---:|:---:|:---:|
| 0.25 | 12.44/12.74 | 3.11/12.74 |
| 1.5 | 74.6/76 | 18.66/76 |

[1] Mean velocity (cm$\cdot$ s$^{-1}$)/Minimum inlet length (mm).

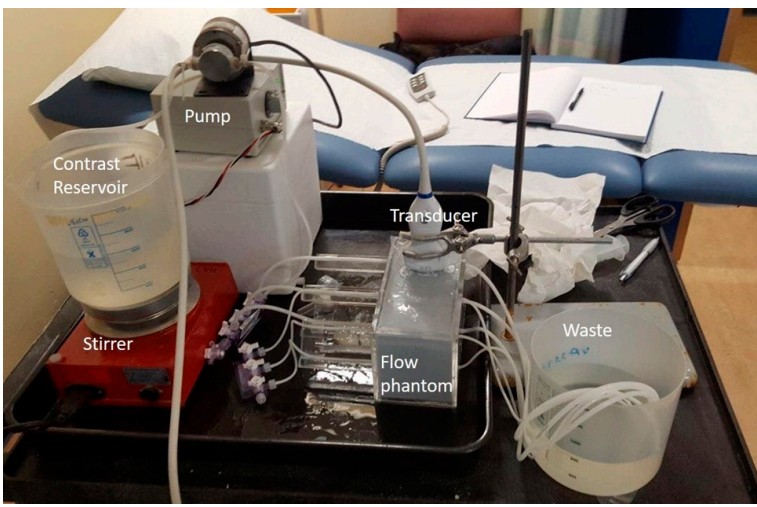

**Figure 1.** Experimental Set-up. Transducer mounted using retort stand on flow phantom. Contrast diluted in reservoir, then pumped into flow phantom.

During the experiments the maintenance fluid was removed from the top of the phantom and the transducers were mounted using a retort stand and clamp so that each tube could be imaged sequentially. The final 0.5 cm of the phantom was filled with water for experiments undertaken with the linear transducers and with gel for the curvilinear probes. Both techniques ensured good acoustical coupling of the transducers with the phantom surface.

A reservoir of degassed water was then pumped sequentially through each of the tubes at two flow settings—0.25 mL s$^{-1}$ and 1.5 mL s$^{-1}$ using a Micropump suction pump

(Micropump, Inc, Vancouver, WA, USA). To determine the mean velocity within each of the pipes the equation of continuity was used such that:

$$Q = v \times A \tag{3}$$

where Q is the flow rate, v is the mean velocity of the fluid and A is the cross-sectional area of the space the fluid is moving through. The mean velocities within each of the tubes are shown in Table 1.

Each scanner was setup in default contrast mode using the same criteria. Acoustic output values were set automatically for all transducers. Depth was adjusted so that each pipe could be visualised in the B-mode scan-plane; the focal point was positioned to below the distal edge of the tube and the gain of the system was adjusted so that the distal edge of the tubing was just visible in the B-mode image when water was circulating through the tube—at almost all depths this resulted in the tube and the TMM not being visible in the contrast-specific imaging mode image (Figure 2).

Adjustments of other controls on the scanner were left to the discretion of the clinician. A solution of SonoVue or DEFINITY® was pumped sequentially through each of the tubes at the two flow settings. Three millilitres of freshly activated SonoVue or 0.3 mL of room-temperature DEFINITY® were diluted in 1 L of deionised water so that there were approximately the same order of magnitude of contrast microbubbles/mL in each solution (Table 2) ($1.68 \times 10^6$ microbubbles/mL SonoVue and $1.92 \times 10^6$ DEFINITY®).

**Table 2.** Contrast agents and physical characteristics of reconstituted microbubbles (from manufacturers' literature).

| Contrast Agent/ Manufacturer | Microbubbles/mL | Mean Diameter Range (μm) | Shell/Gas | Gas Volume (μL/mL) | Clinical Dose |
|---|---|---|---|---|---|
| DEFINITY®/ Lantheus Medical Imaging | $6.4 \times 10^9$ | 1.1–3.3 98% < 10 μm | Lipid/octafluoropropane | 150 | 10 μL/kg |
| SonoVue/ Bracco Diagnostics Inc. | $1.5-5.6 \times 10^8$ | 1.5–2.5 99% < 10 μm | Lipid/sulphur hexafluoride | 8 | 2 mL (cardiac) 2.4 mL (liver) |

The solution was continuously stirred using a magnetic stirrer. If independent gain settings were available in contrast mode, these were set a mid-range values selected by the clinician. After passing through the phantom, the contrast solution was sent to waste. All tubes within the phantom were scanned in contrast-enhanced mode at all three depths and both flow regimes using 4 transducers: a GE Logiq E9 (C2-9) curvilinear probe, a Philips iU22 L9-3 linear array probe, an Esaote MyLAbTwice linear array LA523 (4–13 MHz) and a Fujifilm VisualSonics Vevo3100 MX250 (21 MHz) probe (Table 3).

**Table 3.** Scanner and Transducer combinations used in the study and CSIM employed. Each scanner was setup in default contrast mode.

| Scanner/Probe | Probe Type (Frequency Bandwidth, [Centre Frequency]) | Contrast-Specific Imaging Mode | Acoustic Output Value (Pipe Depth)-Set Automatically |
|---|---|---|---|
| Esaote MyLab Twice Vet | LA523 (2–9 MHz, [4.5 MHz]) | Second harmonic imaging | 40 kPa (all depths) |
| GE Logiq 9 | C2-9 (2.3–8.4 MHz, [4.2 MHz]) | Phase inversion | 0.13 (1 cm); 0.13–0.16 (5 cm) 0.09–0.10 (8 cm) |
| Philips iU22 | L9-3 (3–9 MHz, [4.5 MHz]) | Phase inversion | 0.06 (all depths) |
| Fujifilm Visualsonics Vevo 3100 | MX250 (15–30 MHz, [22 MHz]) | Amplitude modulation | 10% (all depths) |

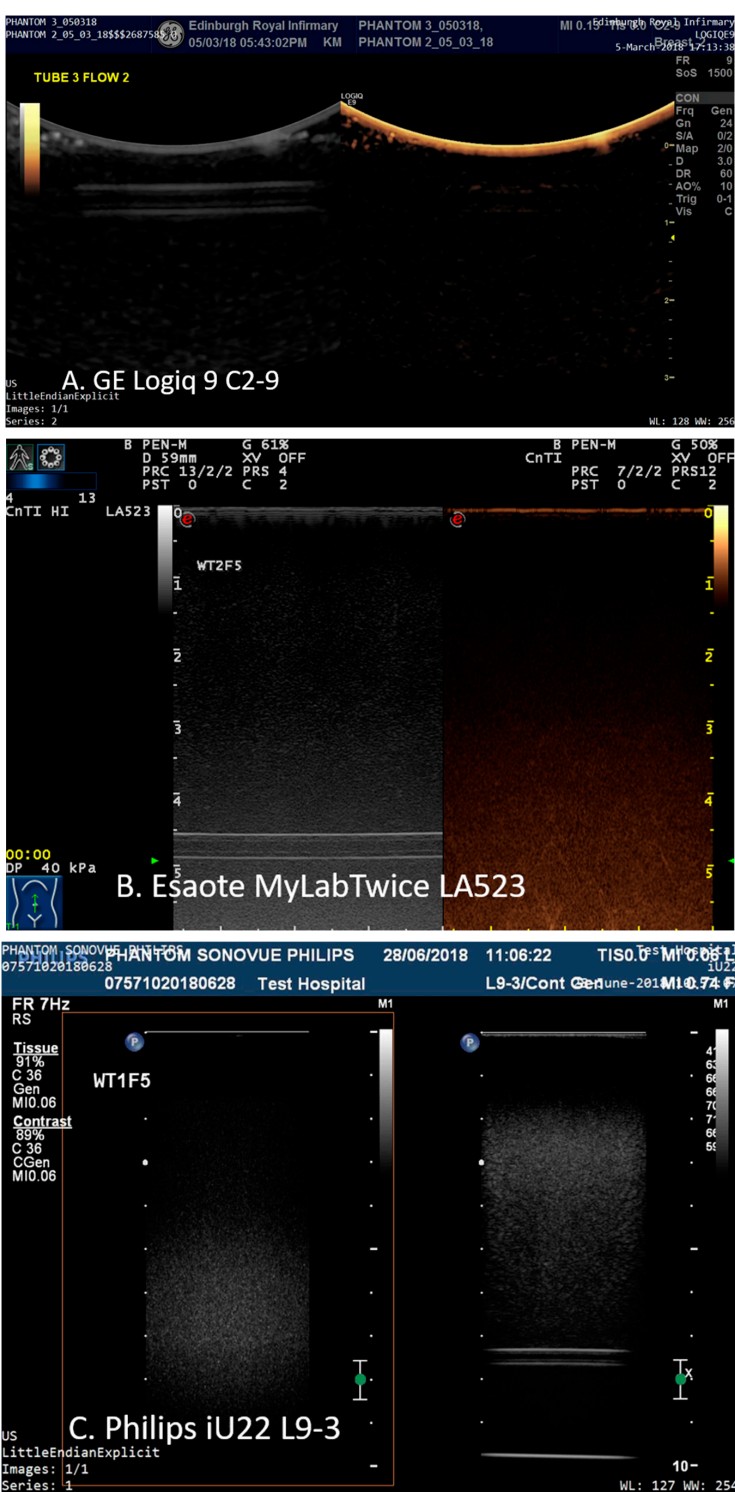

**Figure 2.** Images acquired from 1.6 mm tubing filled with water at (**A**) 1 cm depth imaged using the GE Logiq9 C2-9 transducer; (**B**) 5 cm depth imaged using an Esaote MyLab Twice LA523 and (**C**) 8 cm depth using a Philips iU22 L9-3.

On a separate occasion, the same probes were used again to scan the tubes with the second contrast agent using the identical scanner settings. Images were acquired when contrast enhancement had reached maximum backscatter intensity which was assessed visually by the operator, an experienced radiologist (EQ) with over 15 years' experience of ultrasonic contrast imaging. Images were downloaded in DICOM (Medical Imaging and

Technology Alliance, Arlington, VA, USA) format and subsequently analysed off-line by a medical student (CA).

The images were analysed using Analyze 11.0 (Mayo Clinic Rochester, MN, USA) by CA who was not blinded to the study. Six regions-of-interest (ROIs) were placed on the B-mode images: 3 ROIs proximal to the upper edge of the tubing within the TMM and 3 ROIs centred within the tubing (Figure 3).

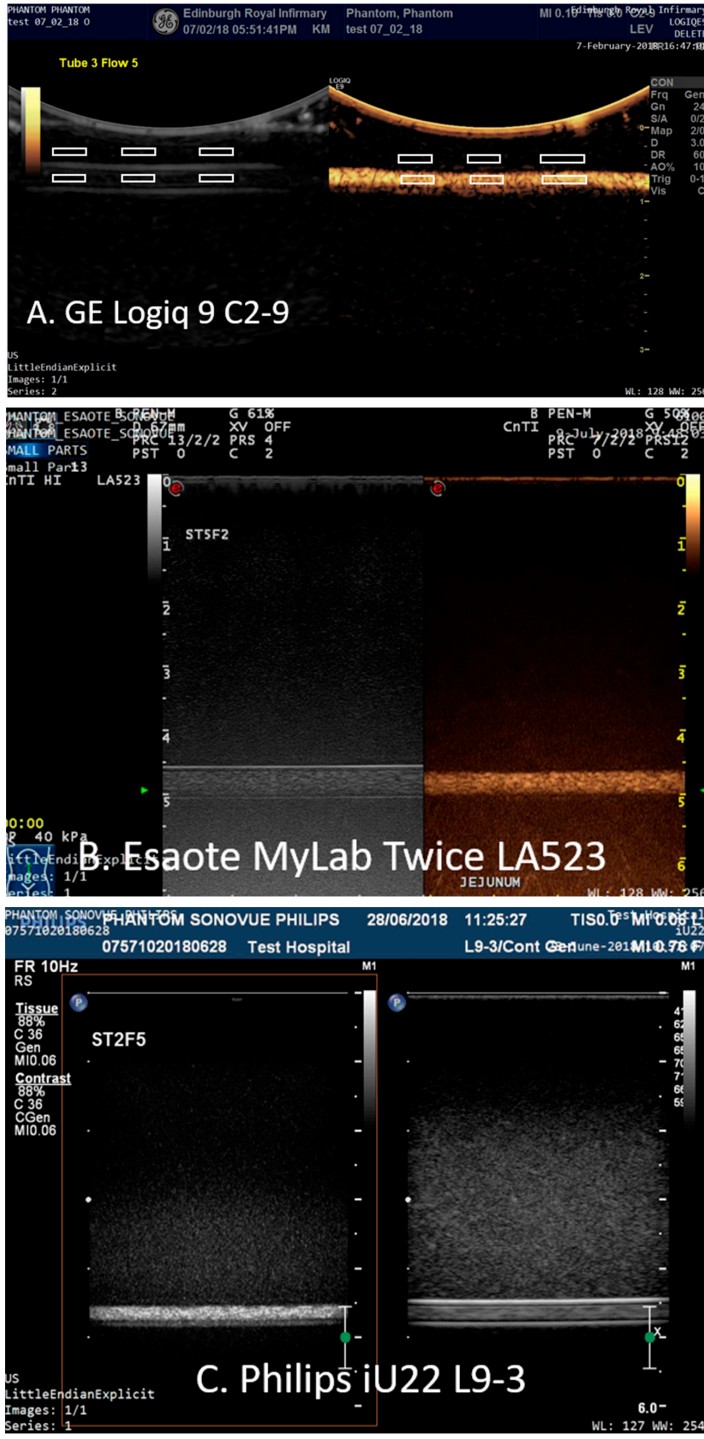

**Figure 3.** Images acquired from 1.6 mm tubing (**A**) 1 cm depth SonoVue-filled tube imaged using the GE Logiq9 C2-9 transducer with ROIs highlighted; (**B**) 5 cm depth SonoVue-filled tube imaged using an Esaote MyLab Twice LA523 and (**C**) 8 cm depth DEFINITY®-filled tube using a Philips iU22 L9-3.

The size of the ROIs was chosen so that there were no reverberations from the tubing within any of the ROIs sited within the tubing and ranged in area from a minimum of 252 pixels (1.6 mm ID tube) to 1725 pixels (3.2 mm ID tubes). The 6 ROIs were then copied to the same positions onto the corresponding contrast-enhanced image and mean intensity of the final 12 ROIs were exported to an Excel spread sheet. The mean intensity from the 3 ROIs sited within each of the tubes was calculated in CSIM and B-mode (BM) and the relative enhancement in contrast-specific imaging mode relative to B-mode (CSIM-BM) was expressed in decibels. Statistical differences were determined using a Mann Whitney non-parametric test to perform a pairwise comparison between the values of CSIM-BM for both contrast agents, tube diameters, depth and flow rates. Significance was set at the 5% level.

### 3. Results

The Fujifilm Visualsonics Vevo 3100 MX250 probe was only able to visualise the contrast-filled superficial tubes in B-mode (Figure 4).

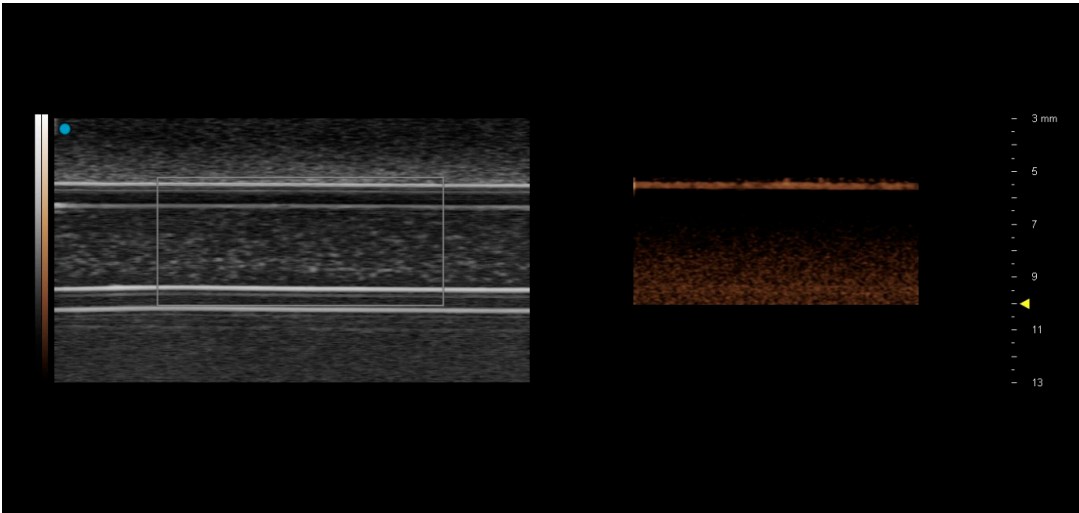

**Figure 4.** Images acquired using the Vevo 3100 MX250 preclinical ultrasound probe from 3.2 mm diameter tubing at 1.0 cm depth. Tubing is filled with SonoVue. Only the superficial layer of the tube can be visualized in the contrast-specific imaging mode image on the RHS.

The matching CSIM image showed a bright line coincident with the upper level of the tubing, with a dark region immediately below the tubing suggestive of shadowing, either by the tubing or by contrast within the upper levels of the tubing. This was consistent for both contrast agents and both flow rates. Consequently, no analysis could be performed on these images.

### 3.1. Contrast Agent

For the Philips iU22 L9-3 probe (Figure 5) and the GE Logiq9 C2-9 (Figure 6) there were no significant differences in the magnitude of the CSIM-BM ratios at each depth, flow rate and pipe diameter for the two agents.

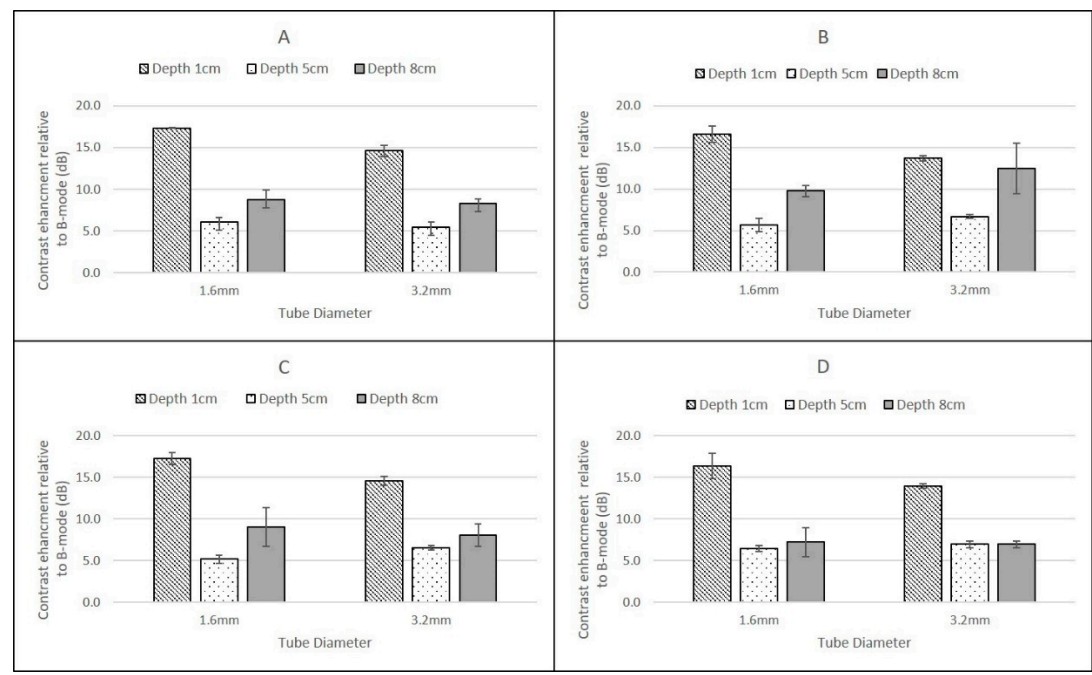

**Figure 5.** Ratio of backscatter signal in contrast-specific imaging mode to B-mode using a Philips iU22 L9-3 probe for (**A**) SonoVue at a flow rate of 0.25 mL/s; (**B**) DEFINITY® at a flow rate of 0.25 mL/s; (**C**) SonoVue at a flow rate of 1.5 mL/s; (**D**) DEFINITY® at a flow rate of 1.5 mL/s. Error bars are standard deviations.

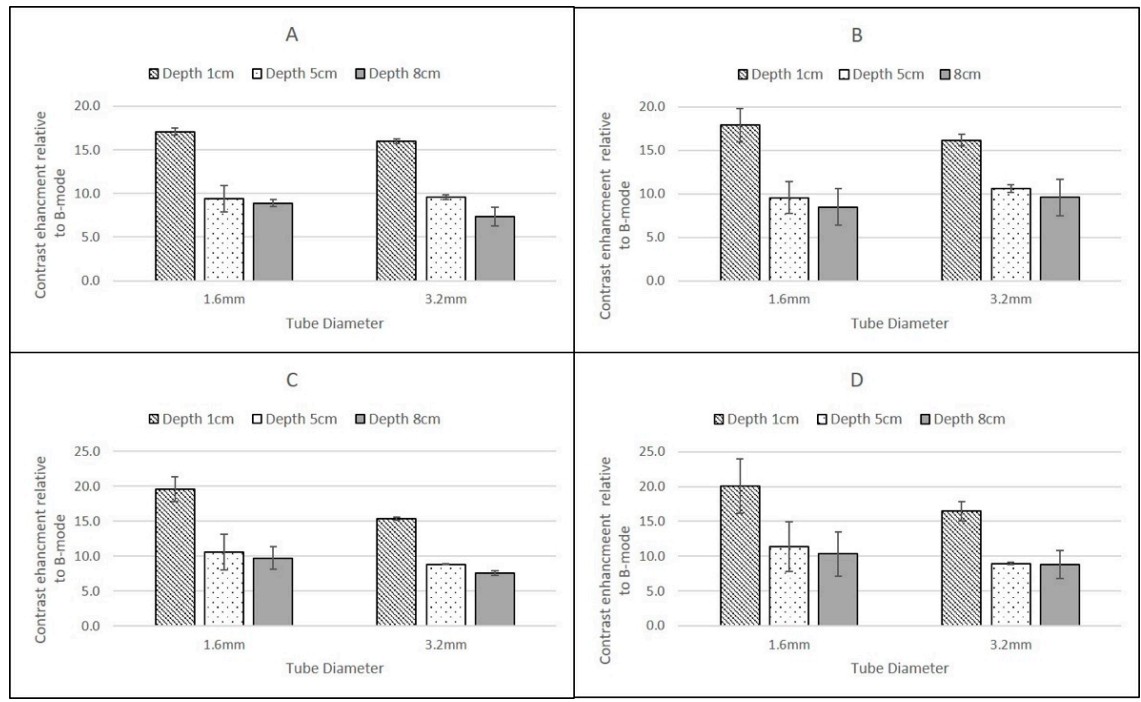

**Figure 6.** Ratio of backscatter signal in contrast-specific imaging mode to B-mode using a GE Logiq9 C2-9 probe for (**A**) SonoVue at a flow rate of 0.25 mL/s; (**B**) DEFINITY® at a flow rate of 0.25 mL/s; (**C**) SonoVue at a flow rate of 1.5 mL/s; (**D**) DEFINITY® at a flow rate of 1.5 mL/s. Error bars are standard deviations.

The Esaote MyLab Twice LA523 probe showed significant differences between agents, with DEFINITY® exhibiting significantly larger CSIM-BM ratios than SonoVue at superficial depths but was undetected at greater depth (Figure 7).

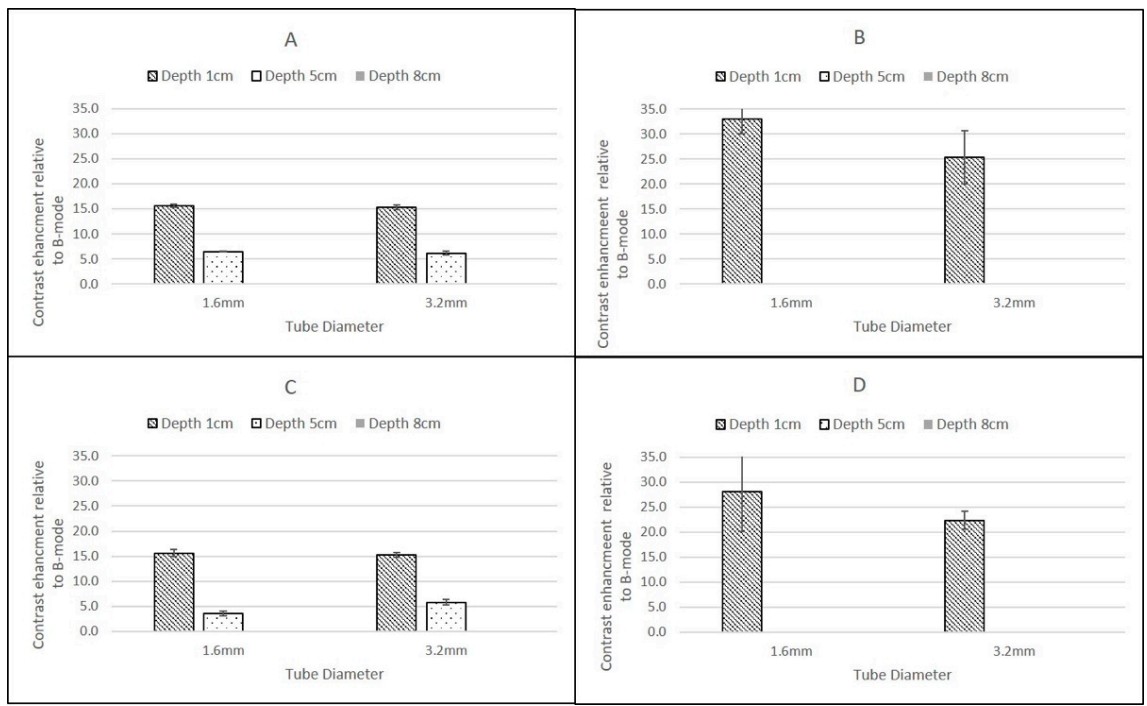

**Figure 7.** Ratio of backscatter signal in contrast-specific imaging mode to B-mode using an Esaote MyLab Twice LA523 probe for (**A**) SonoVue at a flow rate of 0.25 mL/s; (**B**) DEFINITY® at a flow rate of 0.25 mL/s; (**C**) SonoVue at a flow rate of 1.5 mL/s; (**D**) DEFINITY® at a flow rate of 1.5 mL\L/s SonoVue. Note the change in scale on vertical axis compared to Figures 5 and 6. Error bars are standard deviations.

*3.2. Depth*

For the Philips iU22 L9-3 probe (Figure 5), CSIM-BM values at a depth of 1 cm were significantly greater than values obtained at 5 cm and 8 cm depth except for DEFINITY® flowing within the 3.2 mm diameter tube at 8 cm depth (Figure 5). All CSIM-BM ratios for all scanners obtained at 5 cm were smaller in magnitude than those obtained at 8 cm.

For the GE Logiq E9 C2-9 probe, CSIM-BM ratios decreased with increasing depth for both agents, both pipe-diameters and both flow rates with values significantly higher at 1 cm depth compared to 5 cm and 8 cm depth. The magnitude of the CSIM-BM ratios at each depth, flow rate and pipe diameter were not statistically different between the two agents (Figure 5).

Although the Esaote MyLab Twice LA523 probe was only able to detect contrast in the more superficial pipes, the highest CSIM-BM ratios were obtained using this probe when insonating DEFINITY® in both the 1.6 mm and 3.2 mm ID pipes centred at 1 cm depth and at both flow rates. Moreover, this was the only probe which demonstrated significant differences in the magnitude of CSIM-BM ratios between the two contrast agents, with DEFINITY® exhibiting ratios larger than SonoVue values at superficial depths. At both 1 cm and 5 cm depth, the Esaote probe demonstrated similar CSIM-BM, ratios consistent with those obtained for SonoVue from the other scanners (Figure 6). DEFINITY® was undetectable at 5 cm depth.

The GE Logiq E9 C2-9 and Phillips iU22 L9-3 probes demonstrated positive CSIM-BM ratios at all depths, for both flow rates, contrast agents and diameter tubes. The Esaote MyLab Twice LA523 probe demonstrated positive CSIM-BM ratios in the most superficial pipes, centred at 1 cm depth, for both SonoVue and DEFINITY® but only for SonoVue in the 5 cm depth pipes, as DEFINITY® could not be visualised at 5 cm depth nor either agent at 8 cm depth.

### 3.3. Flow Rates

In tubes of the same ID, there was no significant difference in the CSIM-BM ratio between the two flow rates (0.25 mL/s and 1.5 mL/s) at all depths and for both contrast agents, although at the higher flow rate and for deeper tubes there was a tendency for larger standard deviations on the mean values of CSIM-BM. This was true for all scanner–transducer combinations (Figures 5–7).

### 3.4. Diameter of Tubing

For both contrast agents, and for both flow rates and all scanner-transducer combinations, narrow (1.6 mm) pipes centred at a depth of 1 cm demonstrated the largest CSIM-BM ratio. This was greater than larger ID pipes at the same superficial depth and under the same flow conditions and for all other pipes at all depths. This dependence on pipe diameter was not evident for pipes centred at depths of 5 cm and 8 cm and there was no statistically significant difference between pipes of different diameter centred at the same depth (Figures 5–7).

## 4. Discussion

Despite many of the clinical applications of ultrasonic contrast agents being off-label, the clinical utility of ultrasonic contrast agents is well documented in the literature and in European and world guidelines [2]. Low-transmit-power insonation at the resonant frequency of the microbubbles produces non-linear microbubble oscillation resulting in the generation of harmonic frequencies. These harmonic frequencies allow clinical users to distinguish the microbubble signal from native tissue clutter by using specialised contrast-specific ultrasound imaging modes. For these reasons, only US systems equipped with non-linear imaging modes designed for contrast imaging should be used in clinical practice.

From a clinical perspective, contrast-specific imaging modes allow an enhanced differentiation between native tissue and pathologic findings predominantly based on the visual interpretation of the temporal and spatial contrast enhancement provided by microbubble agents. The best contrast resolution is obtained when native tissue does not produce any signal and the only signal displayed by the ultrasound scanners is from the microbubbles. Since many of the clinical ultrasound scanners are very efficient at removing the native tissue signal, in this study it was not possible to use the enhancement from the proximal regions-of-interest within the tissue-mimicking material to normalise the results between scanners as these values tended to zero. In addition, the non-linearity of tissue-mimicking materials has recently been documented and is significantly higher than most soft tissues [16], so results obtained would not be typical of clinical contrast to native tissue ratios observed in diagnostic scanning. For these reasons, we developed the CSIM-BM ratio to determine the sensitivity of contrast-specific imaging modes with reference to the enhancement observed in B-mode imaging at a range of depths and flow rates.

However, although there are many studies reporting comparisons in echogenicity and application of commercial and in-house contrast agents [1,17–19], there are relatively few studies which compare the sensitivity of scanner-specific CSIM of the commercially available contrast agents. The aim of this study was to compare the sensitivity of CSIM from 3 premium clinical, and one preclinical ultrasound scanner under a range of flow conditions typically experienced within humans and small animals. All scanners utilise proprietary techniques for CSIM. The Philips iU22 scanner utilises phase-inversion techniques in its CSIM with MI values in this study remaining at 0.06 for scanning at all depths. The GE Logiq E9 also utilises phase-inversion techniques to isolate the contrast signal. MI values varied with depth, with an MI of 0.13 for imaging the pipes centred at 1 cm depth, 0.16 for imaging the pipes centred at 5 cm and 0.09 for imaging the pipes centred at 8 cm. Esaote MyLab Twice utilises second harmonic imaging and, although no MI values were displayed, the derated pressure was 40 kPa (corresponding to maximum MI values of 0.01 and 0.03, over the 2–9 MHz bandwidth of the transducer). The Vevo3100 preclinical ultrasound scanner uses amplitude modulation CSIM to isolate the microbubble signal. Power output

is defined as 10%, although the magnitudes of acoustic output pressures for the MX250 probe are not publicly available. However, since the MI values are inversely proportional to the square-root of the transmitted frequency, peak negative acoustic pressures up to 0.46 MPa, transmitted at 21 MHz, will still result in MI values less than 0.1. The lack of visualisation of contrast in the tubes in CSIM at the higher frequencies utilised by the Visualsonics FujiFilm MX250 probe is not surprising, and most likely due to a combination of the high attenuation of the C-flex tubing [20] at the transmit frequency of the probe (21 MHz) and that the resonance frequency of the main population of bubbles within both DEFINITY® and SonoVue is significantly removed from this frequency [18,21,22]. Nevertheless, both SonoVue and DEFINITY® were visualised in B-mode and, as such, they have potential application within the high-frequency preclinical imaging laboratory for confirmation of injection sites during ultrasound-guided injections [23].

In the 1.6 mm internal diameter tube, at the low flow setting of 0.25 $cm^3/s$, the contrast solution had a mean velocity of 12.44 cm/s, while at the higher flow setting (1. 5 $cm^3/s$) this corresponded to a mean velocity of 74.6 cm/s. In the larger ID tube, these velocities were 3.11 cm/s and 18.66 cm/s respectively. These ranges of velocities are typical of those experienced clinically in a range of vessels, from larger elastic arteries to smaller arteries and veins. For both contrast agents, and all scanners, the magnitude of the CSIM-BM ratio was independent of flow rate within the phantom, highlighting the sensitivity of all the CSIMs to the wide range of flow conditions and velocities within the body.

The highest CSIM-BM ratios were observed at superficial depths for all scanners, flow rates and contrast agents and for SonoVue these values ranged from $19.61 \pm 1.75$ dB (GE Logiq 9 C2-9, 1.6 mm tube) to $14.59 \pm 0.52$ dB (Philips iU22 L9-3, 3.2 mm tube). For DEFINITY®, although the values for the Philips and GE Logiq 9 CSIM-BM ratios were similar to those obtained from SonoVue, the Esaote MyLab Twice CSIM-BM ratios for DEFINITY® were significantly greater, resulting in a range of CSIM-BM ratios from $33.09 \pm 2.9$ dB (Esaote MyLab Twice LA523, 1.6 mm tube) to $13.68 \pm 0.32$ dB (Philips L9-3 3.2 mm tube). At 8 cm depth, although the tubing could be visualised, no contrast-enhancement was observed in B-mode or CSIM for either agent using the Esaote MyLab Twice LA523 probe, suggesting that the ultrasound beam was too strongly attenuated at the high centre frequency of the probe. Indeed, as this probe is designed for small animal veterinary applications, there is limited clinical requirement for imaging beyond 6 cm depth. For all probes, both agents and both flow settings, for tubes centred at 1 cm depth there was a reduction of CSIM-BM ratios within the larger ID tubes compared to the narrower ID tubes, despite both tubes having identical tube-wall thicknesses of 0.8 mm. This is likely due to attenuation and shadowing caused by contrast enhancement within the proximal layers in the larger ROI within the 3.2 mm tubes, reducing the overall mean enhancement measured compared to the smaller ROIs within the smaller ID tubes. With increasing reduction in beam intensity caused by attenuation of the ultrasound beam as it travelled through the TMM to deeper pipes, this difference between CSIM-BM ratios between pipe diameters was no longer evident. Both the GE Logiq E9 C2-9 probe and the Philips iU22 L9-3 probe gave very similar CSIM-BM ratios at 1 cm and 8 cm depths for both flow rates, contrast agents and tube diameters. At 5 cm depth, for both contrast agents the Philips iU22 L9-3 CSIM-BM ratios were significantly lower than those obtained at 1 cm and 8 cm and lower in magnitude than those obtained by the GE Logiq E9 C2-9 probe. The reasons for this are not clear, but may be due to the positioning of the focus during the Philips measurements. Although the focal region was centred below the distal edge of the tube at 5 cm, the focal range extended into the distal regions of the tube. Since pulse-inversion techniques are dependent on low MI values, the acoustic pressure within the focal region may have been sufficiently high to destroy the contrast in the distal region of the tube reducing the magnitude of the overall CSIM values.

The overall similarity in CSIM-BM ratios from the Philips and GE probes is likely due to both scanners utilising phase-inversion techniques as their CSIM. Hyvelin et al. (2016) compared the contrast enhancement obtained from SonoVue, DEFINITY® and Optison

using two premium ultrasound scanners (GE Logiq E9 and Toshiba Aplio 500) in in vitro studies and found that the properties of DEFINITY® and SonoVue were broadly similar with both superior to Optison. The results obtained in this study from the Philips iU22 and Logiq E9 transducers are consistent with these results demonstrating broadly similar CSIM-BM ratios for both agents at two flow rates and at 1 cm and 8 cm imaging depths. In addition, SonoVue values, obtained using the Esaote scanner at proximal depths are similar to values obtained from the iU22 and Logiq E9 scanner. Although the differences between SonoVue and DEFINITY® using the Esoate scanner are surprising, without more detailed insight into the contrast-specific imaging mode utilised within the scanner it is difficult to determine if these differences are due to specific contrast-agent properties or the CSIM that is utilised within the Esaote scanner.

This study was performed using a range of transducers which were available to the user and also using a flow phantom which generated relatively fast flow rates, typical of those observed in arteries and significantly faster than those observed within capillaries. Future studies will utilise a pump which can generate lower velocities enabling an assessment of the contrast imaging capability of these scanners for capillary-like flow, rather than large vessel flow described in this study. Including tubes at greater and shallower depths will further test the range over which the curvilinear probes and the high-frequency preclinical scanner probes can be tested. In addition, manufacturing wall-less vessels would remove the high intensity echoes generated from the C-flex tubing embedded within the tissue-mimicking material.

## 5. Conclusions

This study utilised the CSIM-BM ratio as a metric to compare the enhancement of contrast-specific imaging techniques relative to their B-mode enhancement using two commercially available contrast agents at a range of different flow values and depths in an in vitro flow phantom. The results demonstrated that both the Philips iU22 and the GE Logiq 9 demonstrated broadly similar enhancement for both agents at 1 cm and 8 cm depths. However, the Esato MyLab Twice gave significantly larger results for Definity compared to Sonovue at superficial depths with neither agent detectable at depth. The Visualsonics FujiFilm preclinical ultrasound scanner Vevo3100 was able to visualise both contrast agents within the superficial tubes in B-mode imaging but not in CSIM, likely due to the increased attenuation of the tubing at high frequency.

The results from this study suggest that both SonoVue and Defintiy exhibit similar flow dynamics and enhancement profiles when imaged using the two premium ultrasound scanners utilising phase-inversion techniques. The Esaote probe, which is utilised for veterinary applications appeared more sensitive to Definity at superficial depths, suggesting that Definity might be useful for superficial imaging. From a clinical perspective this study would suggest that SonoVue and Definity provide similar backscatter from deep structures, such as the abdomen, while it seems that Definity may provide higher contrast gain in superficial applications such as soft tissues or muscles.

**Author Contributions:** Conceptualization, C.M.M. and E.Q.; methodology, C.M.M. and E.Q.; software, C.A.; validation, C.M.M., C.A. and E.Q.; formal analysis, C.A.; investigation, C.M.M. and E.Q.; resources, C.M.M. and E.Q.; data curation, C.M.M. and E.Q.; writing—original draft preparation, C.M.M.; writing—review and editing, E.Q.; visualization, C.M.M. and E.Q.; supervision, C.M.M. and E.Q.; project administration, C.M.M.; funding acquisition, C.M.M. All authors have read and agreed to the published version of the manuscript.

**Funding:** This research was partly funded by Lantheus Medical Imaging, Inc., Grant CG#16016.

**Institutional Review Board Statement:** Not applicable.

**Informed Consent Statement:** Not applicable.

**Data Availability Statement:** The data presented in this study are available on request from the corresponding author. The data are not publicly available due to limited access to Edinburgh repository.

**Conflicts of Interest:** The authors declare no conflict of interest.

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
