# Peer review of "A Comparison of the Sensitivity of Contrast-Specific Imaging Modes on Clinical and Preclinical Ultrasound Scanners"

_tomography, doi:10.3390/tomography8050191_

Round 1

Reviewer 1 Report

Very nice paper on a well conduceted study. I would like to suggest to reduce the lenght of introdution.

The paper has not evident methodology limits.

It adds to the present literature a new research that is  the assessment ofsensitivity of contrast-specific imaging modes on clinical and preclinical ultrasound scannersIt has been well written and provides useful informations.

I consider it substantially appropriate in all the other items you requested.   Consider I was short and straightforward since I followed the scheme provided but I appreciated the paper.

Author Response

Very nice paper on a well conduceted study. I would like to suggest to reduce the lenght of introdution.

R: Thank you. We removed a complete paragraph on introduction to address this appropriate reviewer concern.

The paper has not evident methodology limits. It adds to the present literature a new research that is the assessment of sensitivity of contrast-specific imaging modes on clinical and preclinical ultrasound scanners. It has been well written and provides useful informations. I consider it substantially appropriate in all the other items you requested.  Consider I was short and straightforward since I followed the scheme provided but I appreciated the paper.

Reviewer 2 Report

The authors present an experimental study comparing various ultrasound scanners in contrast-specific imaging mode using two commercial contrast agents. The paper is well written. My minor comments are:

1. The central frequencies used by all scanners should be included in table 3. It appears that the center frequency used has an impact on the results.

2.  Explain if the 3 depth dependent "Acoustic output values" of GE Logiq9 scanner were set  automatically or manually. It appears that depth dependent acoustic output values should be used in Philips iU22 scanner.  

Author Response

The authors present an experimental study comparing various ultrasound scanners in contrast-specific imaging mode using two commercial contrast agents. The paper is well written. My minor comments are:

1. The central frequencies used by all scanners should be included in table 3. It appears that the center frequency used has an impact on the results.

R: Thank you. This is a good point. Centre frequency of the probe has been added to Table 3.

2. Explain if the 3 depth dependent "Acoustic output values" of GE Logiq9 scanner were set automatically or manually. It appears that depth dependent acoustic output values should be used in Philips iU22 scanner.

R: Thank you. Each scanner was setup in default contrast mode using the same criteria. Depth was adjusted so that each pipe could be visualised in the B-mode scan-plane, the focal point was positioned to below the distal edge of the tube and the gain of the system was adjusted so that the distal edge of the tubing was just visible in the B-mode image when water was circulating through the tube – at all depths this resulted in the tube and the TMM not being visible in the contrast-specific imaging mode image. Please see text for changes. In particular please see Table 3 for changes.